# PROSIM in the Cloud: Remote Automation Training Platform with Virtualized Infrastructure

Sabin Rosioru [1,2], Viorel Mihai [1,2], Mihai Neghina [3], Daniel Craciunean [3] and Grigore Stamatescu [2,*]

1   Asti Automation, Calea Plevnei 139, 060011 Bucharest, Romania; sabin.rosioru@astiautomation.com (S.R.);
    viorel.mihai@astiautomation.com (V.M.)
2   Department of Automation and Industrial Informatics, University Politehnica of Bucharest,
    Splaiul Independentei 313, 060042 Bucharest, Romania
3   Department of Electrical Engineering and Computer Science, University "Lucian Blaga" of Sibiu,
    Emil Cioran 4, 550025 Sibiu, Romania; mihai.neghina@ulbsibiu.ro (M.N.);
    daniel.craciunean@ulbsibiu.ro (D.C.)
*   Correspondence: grigore.stamatescu@upb.ro; Tel.: +40-723425323

**Abstract:** The need for high-availability, efficient, training solutions for industrial automation technology is driving the adoption of new tools, software and services supporting the Industry 4.0 paradigm. Specific requirements include vendor agnostic and standards-based programming languages and availability of both physical input–output and industrial communication, while providing a rich and immersive experience through suitable human–machine interface components. The PROSIM platform represents an integrated embedded hardware-software system for programmable logic controller automation training which provides examples in increasing order of complexity, ranging from basic digital logic up to complex manufacturing simulations and process control. A fully software-based solution has been developed in order to complement the laboratory-based system as a training digital twin. We describe the design, implementation and evaluation of the platform as ported onto the HUBCAP collaborative sandboxing environment for virtualized operation. Integration of dedicated communication libraries for Open Platform Communications, Unified Architecture, with simulated programmable logic controllers such as PLCSIM and ARSim is also discussed for fully leveraging new features emerging from increased information and operational technology convergence witnessed in realistic, plant-level, environments. Qualitative and quantitative evaluation in comparison to the physical system while leveraging the resources provided by the cloud environment is carried out.

**Keywords:** industrial automation; training; process simulations; OPC-UA; virtualization; cloud platform; Industry 4.0

## 1. Introduction

Enabling remote automation technology training for higher education and continuing professional education requires suitable hardware–software platforms that leverage recent advances in cyberphysical systems and industrial internet of things. Specific topics include sensor technology, industrial communication, drive systems, programmable logic controllers and human–machine interfaces. Key design objectives are to enable easy knowledge transfer and practical expertise to the trainees, while balancing intuitive operations with the complexities that exist in a realistic plant or factory environment. Automation training is usually carried out in a dedicated, well-equipped laboratory that allows for physical implementation of the solution with failure simulations for increased realism. Recent public health challenges have accelerated the trend towards e-learning and virtual learning solutions, including for hands-on, resource-intensive fields such as industrial automation technology. This situation, overlapping an already-existing underlying trend for digitalization of Industry 4.0 education, has led to hybrid or fully virtual training courses where the student can initially gain expertise on the automation of a continuous process simulation followed by hands-on sessions in the laboratory. This represents, in some cases,

significant time and cost benefits with added environmental advantages stemming from a reduction in the need for national or international travel. Secondary benefits can also be highlighted from the perspective of educational institutions, which are able to limit capital investments in hardware and easily scale a software solution depending on the number of students and the training courses that they follow every year. The long-expected lifespan of hardware equipment in industrial automation systems also supports the importance and interest in high-level proper automation training solutions.

The PROSIM process simulation automation training system has been initially conceived as a practical solution for programmable logic controller (PLC) training that combines the commercial automation hardware with custom software. The concept is vendor agnostic, allowing for PLC hardware from any manufacturer to be used as long as it complies with the required number and type of input–output channels. An early version of the system with firmware-based microcontroller-level simulation is described in [1]. The system contains a Siemens S7-300 family PLC training board, including a power supply, central processing unit (CPU), analog and digital input–output modules, and a demonstration panel connected through physical input–output signal to the PLC. The demonstration panels includes lamps, switches and potentiometers for the interaction with the programmed logic together with detachable masks and a selector switch that allows the selection of the programming task according to the studied documentation. The following PLC programming languages are supported according to the IEC 61131-3: ladder diagram (LAD), statement list (STL) and function block diagram (FBD). An upgraded version (ASID) is presented in [2] where the microcontroller-based solution is replaced by a general-use embedded industrial PC with suitable RS-485 and Modbus data-acquisition modules (DAQ). The user interface is developed using C# programming language using the Microsoft Visual Studio integrated development environment (IDE) and Expression Blend environments. A launcher application allows the selection of and switching between various process simulations whereas the documentation, such as control narratives and i-o lists required by the student for PLC-side control logic implementation, is embedded into the application using contextual menus. A customized example of process simulations running on the ASID version for energy system modelling is carried out by [3]. These include an automatic circuit recloser, a burner system control and microgrid energy management system simulations. One-to-one mapping between the electrical diagrams corresponding to the control scheme and the software implementation is performed. Faults can be simulated as well as parametrization of the time constants corresponding to the physical equipment characteristics for increased realism. The touch panel system allows intuitive interaction by the students and fast visual feedback of the process simulation state. Various modelling languages and tools can be used concurrently for systematic discrete (Grafcet, Petri Nets) or continuous control system design (feedback control using technical computing software such as MATLAB and Simulink). The solution is compatible with both proprietary and open-source solutions through standards-based communication and software libraries for Modbus implementation and additional simulation developments.

Within a recent context of integrating new technologies and models into industrial automation training systems, this article emphasizes the following contributions:

- Design and implementation of a fully software-based, yet realistic, system for remote automation training using simulated process models, interfacing of input–output signals through IIoT-enabled Open Platform Communications-Unified Architecture (OPC-UA) communication and simulated commercial-grade PLC development environment (B&R ARsim);
- Integration of the proposed solution within the HUBCAP collaborative cloud sandboxing environment and highlighting platform features and resource metrics.

The main differentiation of this work in comparison with a considerable body of knowledge concerned with e-learning, virtual and distance learning solutions for university-grade and technical training in automation technology consists in the deep component-level integration which allows full replicability and porting of the implemented routines once

the physical hardware becomes available. Through the functionalities of the HUBCAP cloud environment, inter-sandbox exchanges are possible, which opens up the possibility of extending the PROSIM solution with related models and tools published by compatible third parties, e.g., dynamic discretized plant models with suitable control narratives and io lists to be connected to the virtual PLC using the PROSIM launcher and software libraries.

The rest of the article is structured as follows: Section 2 is dedicated to a review of the state of the art that frames our contributions within the existing scientific and technological developments; Section 3 presents a detailed perspective on the industrial process simulator development and the components that enable a fully virtualized solution in the cloud; Section 4 introduces the HUBCAP sandboxing environment with detailed technical aspects that support the deployment of the PROSIM applications in a virtual machine; Section 5 illustrates, in detail, implementation aspects for representative process simulations, industrial communication primitives and PLC application development along with the operation of the fully software-based solution in the sandboxing environment. Section 6 concludes the article together with lessons learned and future potential developments.

## 2. State of the Art

With reference to the preliminary description of our system from the introduction, a parallel development using a similar concept for a web-based front-end e-learning approach for various systems such as DC-motor and three-tank plant is presented in [4,5]. Detailed mathematical modelling for the plants is described together with their discretized implementation. The control design is initially validated in MATLAB and subsequently ported in a backed web application. The solution is readily accessible via the internet on end-user devices such as desktops, laptops and tablet PCs. Integration with the control hardware, namely a Siemens S7-1200 family PLC is carried out by means of an Ethernet data-acquisition hardware. TCP/IP communication allows the access to the coupler while physical signals such as analog and digital inputs and outputs are conveyed to the PLC system through wires. Control performance is illustrated comparatively between the simulation approach and the e-learning discretized plant model with physical hardware controller using typical metrics such as steady-state error, settling time and overshoot. Additionally, for the more complex three-tank model simulation an enhanced functionality is provided that allows local storage of the experiment files by the students on a dedicated server for evaluation and replicability purposes.

Relevant related works in the field of (virtualized) e-learning and process simulations in real scenarios that also include hardware modules and components are discussed next, to establish the scientific context and relevance of our contributions. In [6], a web-based solution for student training is described, which revolves around a web application together with RESTful services for data and command exchanges. The system is applicable for discrete manufacturing processes such as flexible assembly lines with a modular software architecture. Validation with the real hardware manufacturing stations is achieved through a dedicated S1000 RTU gateway unit. The two versions, connected to the physical line and to the simulator via TCP/IP communication can be operated in parallel in order to compare the accuracy of the simulation and observe the real behavior of the system. Several considerations are discussed from the point of view of the students which enables safe, remote operation of laboratory installations. In [7], the interoperability between industrial-grade automation equipment and low-cost generic purpose hardware is discussed. The authors describe an application that enables PLC code-testing and SCADA application development for students on prototype plants using an Arduino microcontroller module as signal interface. State machines are used to describe the physical process based on which the control logic is subsequently implemented and tested. Modelling and design for AR/VR applications of industrial automation simulations is presented by [8]. The authors acknowledge the emergence of rich, immersive, multimedia technology with direct application in industrial process modelling for digital twin engineering. This in turn requires suitable training methodologies. The main contributions involve the addition of human–robot interaction in the case of collaborative robots, a large set of tasks and a flexible design that allows

the decoupling of the simulator-supporting architecture from the simulation applications. The framework defines modules and their integration through interfaces and associated UML models. A state manager module coordinates the concurrent and sequential operation of various defined tasks and subtasks. Examples are provided in aircraft assembly, high-voltage cell and machinery tool operations. Enabling technologies include JSON for process definitions, C# for software implementation alongside the Unity3D engine for environment generation, visualization and interaction. Deployment is envisioned through a client–server architecture where the server runs the prespecified models and multiple clients can connect to it and visualize, operate and parametrize the simulation. The key role of OPC communication for linking up sensors, controllers and instrumentations is further emphasized by [9]. Several use cases with dual applicability in both R&D and educational activities are described for automation of energy systems, networked industrial laboratory and educational hardware-in-the-loop platforms. Standardized communication buses enable integration between various field equipment and controllers through software implementation of the OPC server agents using dedicated environments such as WinCC flexible or NI OPC. Wider availability of OPC clients is observed in both Matlab and LabVIEW Datalogging and Supervisory Control modules. A key idea of the study is the reusable nature of the laboratory architecture with various technologies and for various practical applications that replicate and extend existing hardware platforms for learning and experimentation. In [10], the PLC is defined as a key smart service provider for Industry 4.0 production systems for supporting smart industrial control services (SICS).

New concepts that emphasize the importance for automation and PLC training in the context of capacity building for higher education are detailed in [11]. A conceptual benchmark system is presented which integrates Eaton–Moeller software and hardware with suitable input–output modules and devices on a single physical board. The goal is to facilitate teaching of ladder diagram, function block diagram and GRAFCET diagrams. In this case, the system requires extension modules in order to be able to realistically simulate industry processes beyond the learning of the PLC program development tasks. A semivirtual simulation training platform based on PLC hardware is presented by [12]. The authors describe the connection of a previous-generation Siemens S7-200 PLC system via a serial RS232 to a host system HMI for enabling intuitive training. Several simulation applications have been developed and documented, including the program design, i/o list and external wiring diagram for traffic light control, knowledge competition and a material car. It is concluded that such a solution increases the student interest in the content taught and mitigates the limitations due to available hardware stands for process models in the laboratory. An educational perspective of using such automation simulators in online teaching is discussed in [13]. The training set-up includes a virtual Siemens S7-1200 PLC and TIA Portal software which is connected to a Factory I/O virtual environment for modelling the industrial processes from a mechatronics perspective. The authors emphasize the importance of detailed study guides and step-by-step instructions to support the students and trainees during remote teaching. An important observation is to focus on building associations between the rich graphical simulations and the expected behavior of the real physical system with added complexity and uncertainty. As an alternative, a low-cost solution that enables remote access to existing laboratory systems is discussed along with practical insights in [14]. The key elements that assure continuity of PLC training include a scheduling system, remote desktop access, a graphical user interface and a microcontroller board. The PLC unit controls two physical experiments in this case, a conveyor system and a sorting machine. Visual feedback of the functionality and validation of correct operation and/or malfunctions is collected through a webcam. The developed Arduino-based interface conveys the GUI commands from the student to actions on the physical platforms, which are also interfaced to the PLC. In such way remote operation of the plant simulation is achieved. An overarching framework that supports the integration of such disparate platforms and systems, currently addressing local needs for automation and PLC training is introduced in [15]. Various typical PLC training assignments are illustrated that can form a common content for a skills-based training curriculum.

A separate but complementary field of study is dedicated to large-scale process simulations in chemical engineering. In these scenarios, usually specialized, domain-specific software tools are coupled with more generic technical computing environments through well-defined communication primitives. The domain-specific tools run the complex process models and generate the simulated process data, which is transmitted to an application in the external computing environment to be interpreted and serve as decision support for control actions. One example is presented in [16] for the coupling between Aspen HYSYS and MATLAB over OPC communication. The value proposal is supported by the fact that chemical process automation involves significant life, economic and environmental risks such that enabling simulated control for the initial design and evaluation of certain risk patterns and malfunction scenarios can help reduce such risks in the engineering phase.

## 3. PRO-CPS Approach for Industrial Process Simulator Virtualization

Three generations of the process simulator SIMED/ASID/PROSIM V+ are briefly introduced in this section to provide a perspective regarding the evolution of the system towards the current cloud-based implementation. The physical realizations of these systems are illustrated in chronological order in Figure 1.

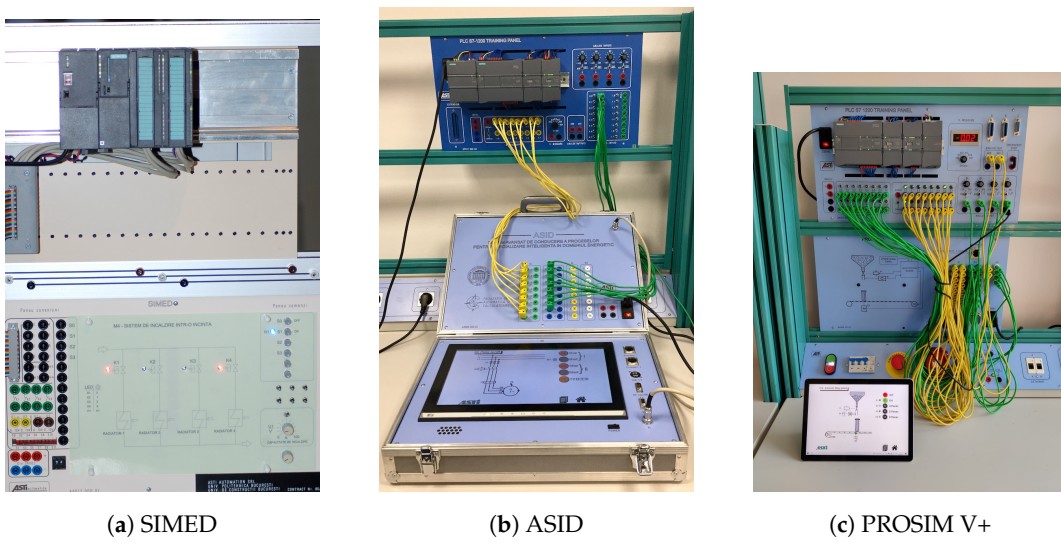

(**a**) SIMED      (**b**) ASID      (**c**) PROSIM V+

**Figure 1.** Three generations of the process simulator for industrial automation.

The earliest version of the concept is represented by the SIMED modular process simulator composed of a demonstration board and a PLC system. The demonstration board is a microcontroller-based embedded system that hardcodes the process logic of the supported process simulations (20) and provides the user a dedicated button to switch the number of the simulation in order to match the existing experiment. The user also has to fit a custom process diagram on the demonstration panel that matches the relevant LED indicator lamps and buttons that are used to physically monitor and interact with the active process simulation. A dense ribbon cable connects the demonstration panel to the PLC unit. In this generation, a Siemens S7-300 family CPU with appropriate input and output modules has been used. Development and implementation of the control logic is carried out using the STEP7 development software according to listed process control narratives and variable lists.

The original version of the integrated training process simulation for industrial automation, ASID (PROSIM T+), is a cyber–physical system by design and leverages the latest advances in embedded hardware and software for developing robust industrial automation training technology. It includes an industrial embedded PC with Intel i3/i5 cpu, 4 GB of RAM and fast SSD storage unit coupled to a high quality 15″ touchscreen display. The data acquisition subsystem provides 6 AI/4 AO 0–10 V and 16 DI/DO 0–24 V channels as modules on Modbus TCP/IP-Ethernet and Modbus RTU-RS485 bus. Applications are de-

veloped using the C# programming language in self-contained Virtual Studio IDE projects. The software platform is based on the Microsoft Windows 10 IoT operating system while 2D/3D process simulations can be developed in multiple frameworks, e.g., Unity [17], Matlab [18], CIROS [19] and deployed on ASID. This generation also increases the number of available simulations to 32. As the OPC-UA protocol can also been supported under Linux-based systems, this can extend the applicability of the solution to a wider class of computing systems.

PROSIM V+ unbundles the three core parts: control hardware, data acquisition interface and process model libraries software as customized solutions. The existing implementation of the PROSIM V+ at the start of the project included 30+ process simulations in increasing order of complexity, as follows: logic basic functions, digital function modules, motor on/off, reversing contactor, star-delta connection, reversing star-delta connection, pendulum table, Dahlander circuit, motor with two separate windings, wound-rotor motor, conveyor belt system, reactive power compensator, heating control, running light, automatic filling system, tank system, fan control, traffic lights for road works, traffic lights, goods lift, coal grinder, embossing machine, collecting belt conveyor, conveyor-charging system, silo control, reactor, pump control, wastewater pump system, monitoring of three pumps, pump system—pressure, drinks machine, sequence control. With the newly developed simulation, the total number of available process simulations has reached 40.

A summary of the hardware, software and functional characteristics of the three generations: SIMED (2008), ASID (2016) and PROSIM V+ (2020), is highlighted in Table 1. With regard to the shortcomings of the previous versions that have driven the need for the current iteration of the process simulator, together with technology advances that enable new functionalities and features, these include: limitation of SIMED to firmware-based simulations (impossibility of easily uploading and extending the system via software updates) with physical process masks for interfacing with the process and lack of industrial communication; built-in touchscreen of the ASID generation that increases the cost, weight and frailness of the hardware equipment in a context where the student and professors should be enabled to use their existing devices and given a choice between a local desktop, tablet or web interface. The new improvements mitigate these shortcomings and provide a more versatile platform which, as a virtualized solution for automation training, offers increased flexibility, lower cost and ubiquitous access to the process simulations and the full PLC application development environment, supported through a wider range of hardware equipment and features.

**Table 1.** Comparative summary of the process simulator characteristics.

|  | SIMED | ASID | PROSIM V+ |
|---|---|---|---|
| **Hardware Platform** | Microcontroller-based embedded system | All-in-one x86/x64 Industrial PC | Embedded PC board |
| **Software Platform** | Custom firmware with hard-coded simulations | C# .NET Applications Windows 10 IoT | C# .NET Applications Windows 10 IoT |
| **User Interface** | Lamps, buttons, switches and user-replaceable process synoptic diagram masks | Built-in touchscreen | Web Tablet PC Smartphone |
| **Communication** | n/a | Modbus TCP/IP Modbus RTU | Modbus TCP/IP Modbus RTU |
| **Control Unit** | Siemens S7-300 | Siemens S7-1200 | Siemens S7-1200 or B&R X20 family |

We envision PROSIM as a family of products and services: stand-alone training unit with embedded software and on board analog/digital i/o (ASID), decoupled software solution running on any Windows platform and dedicated i/o (PROSIM V+) towards

the fully virtualized solution with OPC-UA data communication between the process simulations and simulated PLCs (PROSIM Cloud, detailed in this article). On the services side, the provider and the end user can deliver specialized PLC programming courses to end-users and developers in the process industries and, for larger customers, develop customized process simulations according to their specific equipment/plant. The value proposition lays in the acceleration of knowledge and practical skills acquisition for students, technicians and automation engineers. The technology-agnostic PROSIM platform reduces the need to invest in provider-specific training solutions from the equipment manufacturer. The diversified range of process simulations, in increasing order of complexity from basic digital logic to discrete manufacturing and continuous plant models for the process industries, including a digital-twin model of a control engineering platform, assures step-by-step skills development. Advanced concepts such as mathematical plant modelling with PID control and industrial communication over OPC-UA middleware are exposed to the end user in a packaged form alongside realistic examples of industry processes. All the simulations are documented through the control narrative, functional description and i/o list and are provided with an instructor handbook with hints and solutions for the automation tasks.

The reference approach for a virtualized solution based on this description (main questions relate to what is virtualized and how) is presented in the block diagram from Figure 2. Existing components are marked in green and include the industrial PLC system which supports both industrial i/o interfacing and/or industrial communication using standardized protocols: Modbus TCP, OPC-UA and MQTT to dedicated data acquisition subsystems. These provide a number of digital and analog inputs and outputs which also represent a limitation in the design and implementation of the physical simulator as more complex simulation, making use of over 16 digital input/output channels or 6 analog inputs or 4 analog outputs are not supported. An alternative can be implemented in a hybrid manner where part of the input and output signals and additional signals are relayed via communication protocols between the industrial PLC and the process simulations. The process simulations encode the process models for various typical applications in industry with a one-to-one mapping between the process variables and control signals on both sides.

The components marked in yellow represent new developments. These start with the dedicated package of add-on simulations for the process simulations based on user/adopter requirements in this field. These include mainly flow and thermal plant models in various industries such as food and beverages, chemicals and healthcare. A special digital-twin-type model is also included for the ASTANK2 control engineering platform [20], which is a two-tank system for the experimental study of level, flow and temperature modelling and control. In this particular case, the .NET simulation application is replaced by an executable virtual instrument in which the system model is described either through differential equations or transfer functions and the respective control and output variables are linked to the Modbus registers or the OPC-UA tags. Further on the behavior of the PROSIM solution is similar to with the rest of the simulations. Virtualization of the industrial PLC control unit is carried out using manufacturer-specific software modules that allow the emulation of the respective automation hardware structures while running the same PLC programming projects. In our case, the first option is represented by the Siemens PLCSIM [21] component which provides tight integration with the automation projects developed in the Step7 TIA Portal development environment. This implements a virtual controller which behaves in similar fashion to the physical unit. An alternative, which is illustrated in this article, is the use of the Automation Runtime Simulation (ARSim) component provided in the Automation Studio developments environment supplied by the manufacturer B&R. In this situation, the OPC-UA server functionality is built into the virtual PLC which offers a significant advantage. Finally the packaged simulations, alongside the automation projects built as stand-alone executables including all necessary dependencies are uploaded and tested in a virtual machine provisioned on the HUBCAP cloud platform.

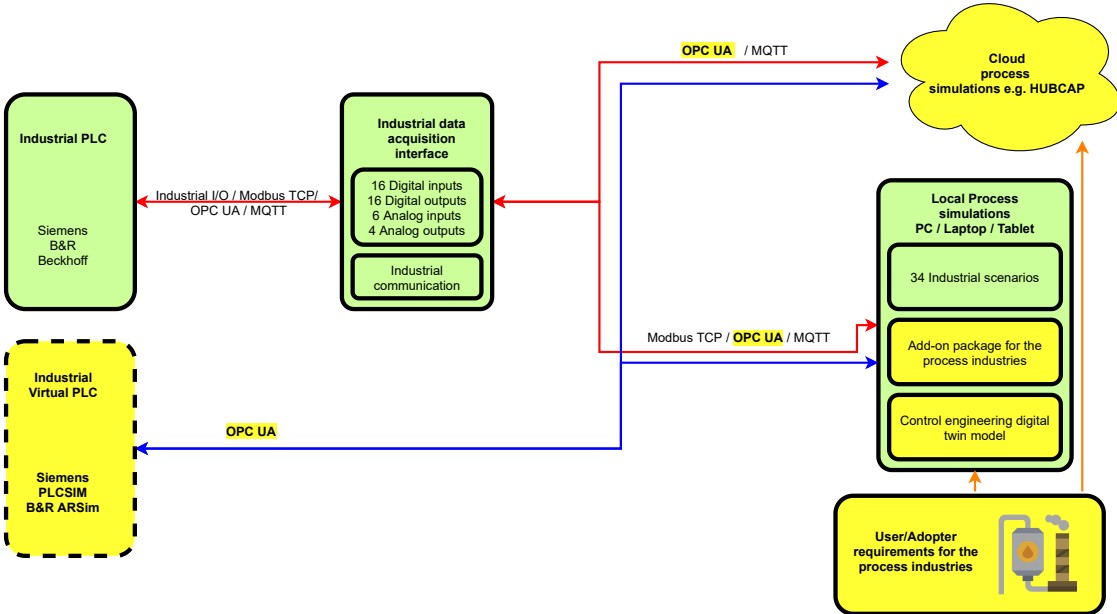

**Figure 2.** PROSIM Cyber–physical platform for application development and training in the process industries (PRO-CPS).

This type of virtualized solution can support new paradigms associated with decentralized and distributed systems over a communication mesh network. We thus also consider enabling solutions for IIoT such as OPC-UA and multiagent systems as [22]. An open-source distributed middleware can be used between the control device and the simulations. The middleware can link together physical and virtual control units which cooperate for control of a single process simulation. Conversely, the simulations can represent parts of a distributed or large-scale system with interlinking process variables communicated over the communication mesh network. One example has been provided in [23] for a laboratory motion-control application with distributed edge devices both industrial-grade, S7-1200 PLC and commodity, Raspberry Pi. Various tools are available to monitor and debug network traffic, either specialized such as UAExpert [24] or general purpose with dedicated profiles for industrial communication protocols such as Wireshark [25] and control applications at runtime. OPC-UA also allows real time operation in the case of time-sensitive applications [26]. In an automation-training context this can lead to quantifying the cycle time and latencies associated to data acquisition and control loops.

In summary, the remote automation-training solution over virtualized infrastructure is achieved by building the automation projects as stand-alone executable files with simulated PLC hardware. The communication between the simulated PLC and the process models is implemented in a server–client manner using the onboard OPC server of the PLC and suitable software libraries on the application side for the OPC client functionality. The supporting platform offers the necessary operating system support (Microsoft Windows Server 2019 in our case) and various features for publishing and controlling user access to the solution. In an extended scenario, data exchange between heterogeneous virtual machines is also possible, e.g., interoperability of process models with a single control application running on one of the VMs.

## 4. HUBCAP Sandboxing Environment

As preliminary, we introduce Model-Based Design (MBD) as a key paradigm for the implementation and management of continuous and discrete industrial processes. MBD [27] is defined as an engineering method that allows the integrated and standardized design of complex systems, such as industrial equipment, automotive and aerospace applications. It uses a combination of mathematical modelling, e.g., differential equations that describe high-order system dynamics with notation conventions that allow standardized

visualization and information exchanges. The virtual representations of physical processes, achieved using MBD, can be operated in a stand-alone manner to visualize plant behavior and test control algorithms, or, as online digital twins, in parallel with the physical process in order to further refine, tune and optimize such algorithms.

Using MBD as fundamental paradigm and enabling methodology, this section is dedicated to the presentation of the Digital Innovation HUBs and Collaborative Platform for Cyber–Physical Systems (HUBCAP) [28] concept, architecture, reusable blocks and integration in the sandboxing environment. HUBCAP has created an open and flexible cloud framework for model-based design assets and related services to be shared, experimented and collaboratively innovated upon. Currently, MBD models are accessible to large industrial players, such as in the aerospace industry, whereas the investment necessary for development, expertise training or deployment may become prohibitive for small and medium enterprises. The platform allows testing and trying out in safe virtual environments, while also providing complete capabilities for collaborations between providers and potential users or for presentation purposes. As an integrated environment, the HUBCAP collaboration platform consists of three main systems: the knowledge-management system, the catalogues-management system, and the sandboxing system, as shown in Figure 3.

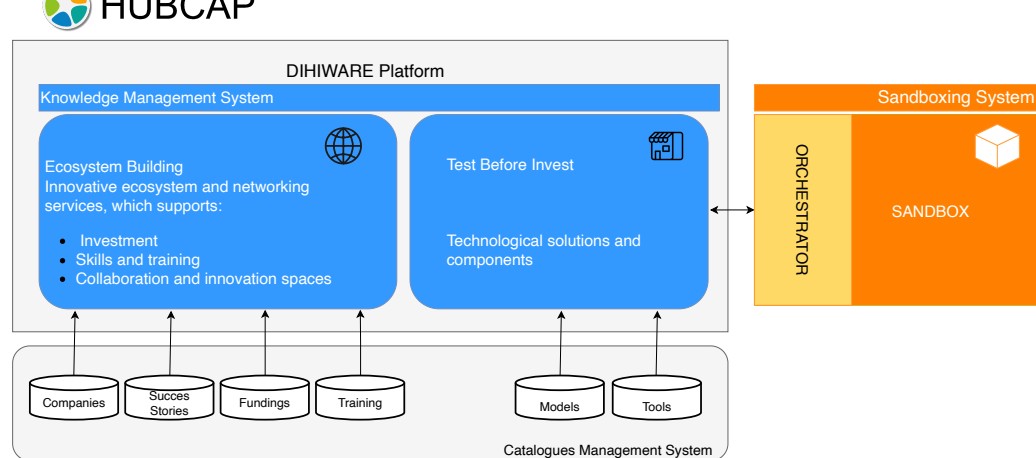

**Figure 3.** HUBCAP Platform Architecture.

The knowledge- and catalogues-management systems are heavily based on the DIHIWARE platform, developed as a part of the MIDIH project within HORIZON 2020 [29]. Envisioned as a middleware, the sandboxing environment has the distinct capabilities of provisioning ready-to-access repositories for both MBD tools and models, as well as executing multiple instances safely, concurrently, and independently from each other. As shown in Figure 4, each sandbox instance is a set of virtual machines (VMk) connected through a shared storage and a dedicated and isolated network for interactions. Within a sandbox, the tools in the HUBCAP platform can be interacted with via a user interface, command line or APIs, or even from outside the sandbox as depicted in Figure 5, but with a limitation: a tool running inside the sandbox can interact with an external system only if the connection is established from (or starts from) the sandbox, i.e., client runs in sandbox, server is outside the platform. The sandbox cannot yet expose APIs to the external world.

The implementation related to the sandbox middleware is done in such a way that the system is kept clean, ensuring that a large number of users have access to the platform simultaneously. The number of virtual machines, models (excluding duplicates) and other resources available for one configuration of the sandbox is balanced dynamically, ensuring that each user has an efficient and productive session. The experimentation functionality of the platform is crucial for hands-on testing, as in the case of the mentality of testing-before-investing, or when models are showcased for promoting or teaching purposes. The complementary functionality allows providers to add new or improved tools and models into the platform, making them visible and testable by potential customers

and collaborators. This provisioning capability is essential for the organic growth and sustainability of the HUBCAP platform, which aspires to become a one-stop-shop for small and medium enterprises wanting to embrace digital innovation through the MBD technology for cyber–physical systems.

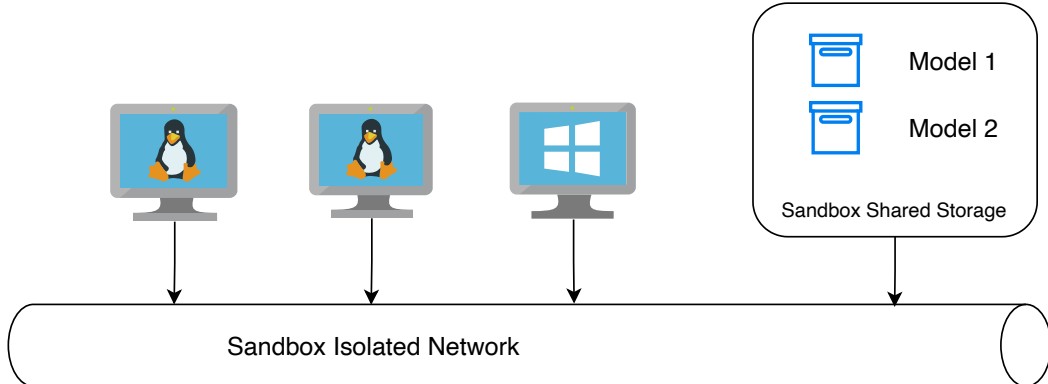

**Figure 4.** A sandbox with a basic OS and two tools. On the right: the sandbox shared storage.

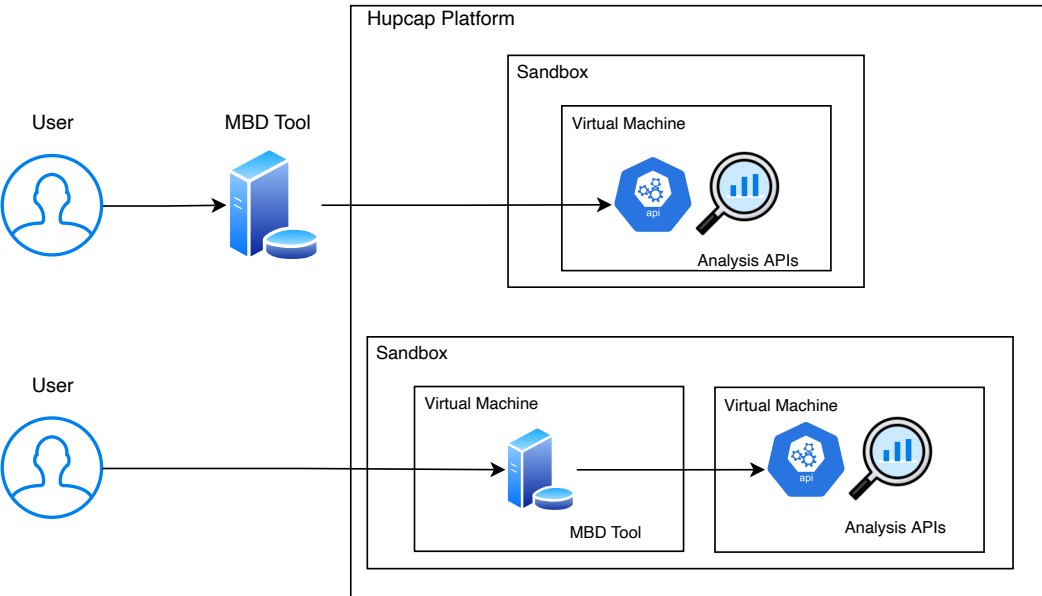

**Figure 5.** Platform architecture: interaction scenario.

While building out a broad repository of resources for MBD testing, the HUBCAP platform differentiates between two fundamental resource types: models and tools. A model represents a building block that encodes the dynamics of a physical system using readily available notations and software. A tool is built around one or multiple models and can be seen as a stand-alone software application. Tools also have the ability to communicate with one another inside the HUBCAP environment and the creator of the tool can establish particular licensing options for the end-user such as software-as-a-service agreements.

To exemplify this classification, one relevant system is the INTO-CPS three tank model [30], that can be publicly accessed in the HUBCAP sandboxing environment. It represents two connected water tanks, with an inflow of water in the first one and the outflow of the second one is collected by a third tank. A typical control application can use this model to assure that the reference level is followed in one of the tanks considering the system behavior and performance requirements. Conversely, in [31] a smart energy tool is described in detail. The application helps users to quantitatively evaluate investment related metrics for energy systems development with suitable visualization, KPI and data

management options. The authors also emphasize cloud-deployment aspects of the tool with regard to sandbox setup, user authentication and performance tests.

The relevance to cyber-manufacturing of the HUBCAP environment is argumented in [32]. The main features that can be used by companies in this field relate to plant simulation, rich 3D visualization and future AR/VR enable maintenance and remote operation applications. The key technical features that support the model and tool integration are listed as the network file system (NFS) and virtual networks (VLAN) inside the environment to isolate sandboxes within suitable integration and communication policies. Consideration of security aspects [33] at the intersection of IoT, CPS and Industry 4.0 systems is vital for the design of robust and reliable solutions. Several types are mitigated by leveraging the HUBCAP platform such as preventing control over communication, data-packet infection and flooding attacks. The virtualization approach can be used also to model and simulate attacks on physical devices and embed measures to protect them in subsequent real implementations.

## 5. Results

The key contributions of this section are listed below:

- Description of three new process simulations including technical details and logic diagrams;
- Implementation details regarding OPC-UA communication;
- Illustration of the solution running within the HUBCAP sandbox.

The main drive of developing new training solutions for engineers comes from the need to understand and be able to maneuver precisely around the real industrial process. Thus, the following section contains the description of three industrial applications and their integration into the HUPCAP platform. Following a point of view at the intersection of industrial/OT and IT domains, the simulations are developed for the Windows operating system using the C# programming language for the frontend and backend. Communication between the software simulation and the simulated PLC unit is done via local OPC-UA tags for which the behavior can be observed or debugged using suitable network monitoring instruments [34].

### 5.1. Process Simulations

For the purpose of the application we illustrate the implementation of three process simulations: pumping station, heat exchanger and flow meter.

The first of the simulations, the Pumping Station in Figure 6, represents a commonly industrial autonomous system equipped with various types of pumps and sensors. The equipment is used to provide fresh and grey water specific to the application area of usage. The simulation consists of two independent systems able to store a quantity of water and the use it for different procedures. The first system must ensure a steady supply of liquid in case of fire emergency and the second system must ensure the water for domestic consumption. The process is equipped with six pumps, one used for filling the tanks, three used by the emergency fire application and another two used for the household consumption. Each of the six pumps can be simulated as faulty by the press of a button; thus, the difficulty of the simulation is increased, and the degree of realism is better molded into a real application, where multiple risks can materialize. Besides the actuators, the simulation is enhanced using digital and analog sensors.

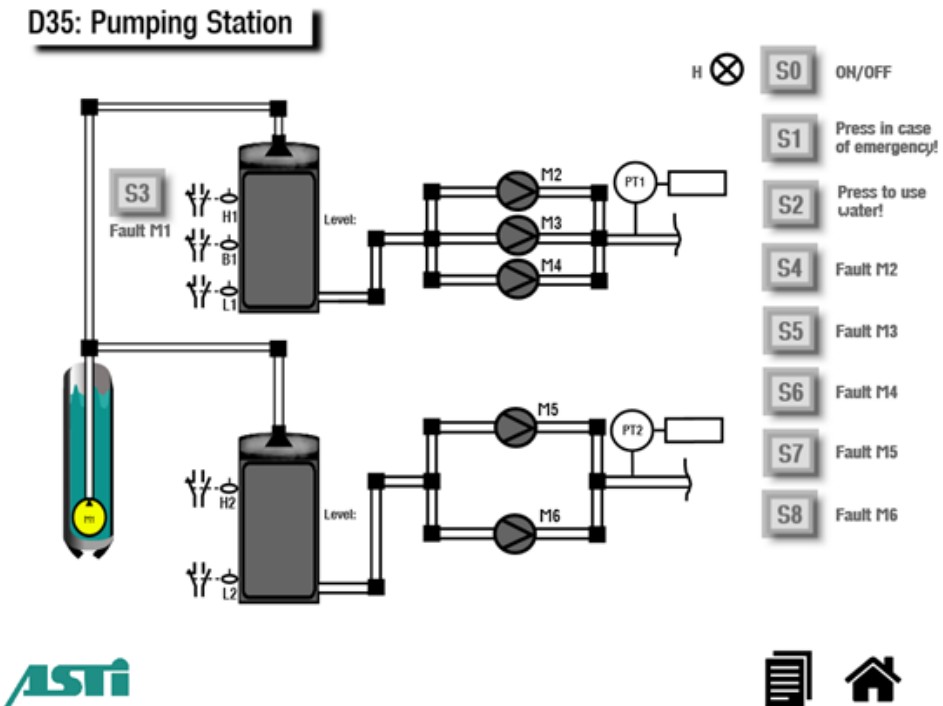

**Figure 6.** Pumping station simulation environment.

*Functional description*: The S0 switch will power ON the system. The power ON/OFF status is signalled by the lamp H1. If the system is powered ON, the control system must control the M1-M6 pumps to supply water on two separate piping networks. The M1 pump must be turned on if the water level in Tank 1 is below the B1 sensor or if the water level in Tank 2 is below L2. The M1 pump will be turned OFF if both tanks are filled, i.e., level H1 and H2 are reached. If the water level in Tank 1 is bellow sensor B1, the PLC must generate an alarm because the minimum water reserve was reached. Such a fault can be simulated by pressing the S3 button. The M2-M3-M4 pumps should ensure a minimum pressure of 3 bar on the emergency piping system. The pressure is measured by the PT1. The transducer measurement range is 0–10 bar. When the pressure drops below 3 bar, equivalent to someone pressing the fire emergency button, the PLC must start two pumps out of three. The selection is made according to the use time, that is, the pumps with minimum use time are chosen. When the pressure reaches 6 bar, the pumps are stopped. If no water remains in Tank 1, determined by the water measurement below the L1 sensor, all pumps must be stopped. If a pump has a fault, before starting or in running, the PLC should start the other pump. The M5 and M6 pumps should ensure a minimum pressure of 3 bar on the second piping system. The pressure is measured by PT2, with the transducer measurement range is 0–10 bar. When the pressure drops below 3 bar corresponding to an active water consumer, the PLC must start one pump out of two. The selection is made according to the use time, i.e., the pump with minimum use time is chosen. When the pressure reaches 6 bar, the pump is stopped. If no water remains in Tank 2, corresponding to the water below the L2 sensor for at least one second, all pumps must be stopped. The water usage is simulated by pressing the S2 button. If a pump has a fault, either before starting or in run mode, the PLC should start the other pump.

The system is programmed to follow three functionality threads as detailed in the UML diagram from Figure 7. If all the safety parameters are in good condition and if the system has been started, then the software will begin the three loops. The first one will handle the water level in the tanks, water needed to ensure a proper functionality. The second one will deal with the Emergency Case circuit. When the pressure will drop below a certain threshold, two out of the three pumps will start running based on the total

function time, in order to reduce the wear of the equipment. The last thread will handle the Use-of-Water circuit, which has the same structure as the previous one.

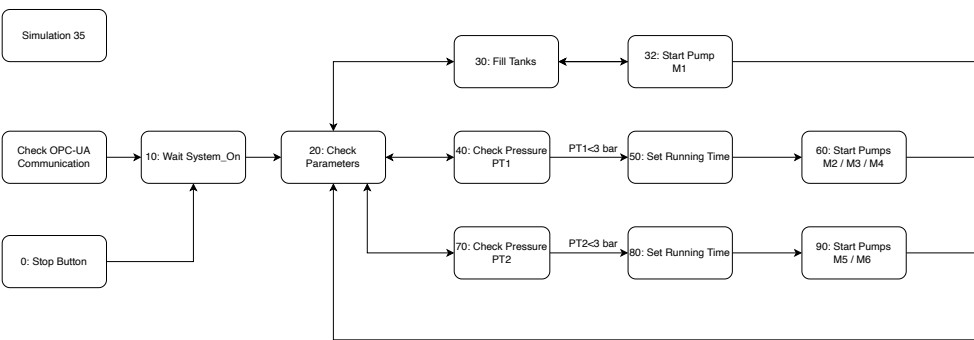

**Figure 7.** UML diagram—pumping station.

Heat exchangers allow end users to reduce their carbon footprint by recovering the waste heat of the generated emissions produced in industrial process. Thus, the energy cost is reduced, and the air exhausted into the ecosystem can be filtered of dangerous chemicals at an intermediary stage. The simulation frontend user interface is shown in Figure 8.

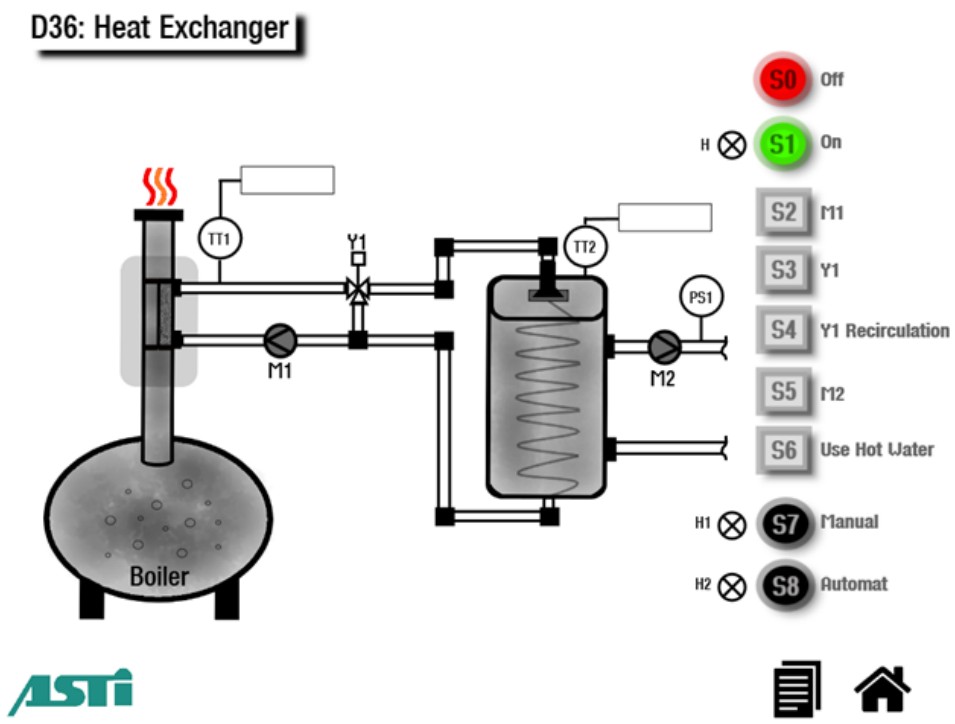

**Figure 8.** Heat exchanger simulation environment.

*Functional description 1 (manual mode):* The hot air resulting from the burning of the raw material reaches the chimney where the SP1 bypass is located. The installation is started by pressing button S1 and stopped by pressing button S0. The lamp H indicates operation. Pressing the S7 button will set the control system into Manual mode. When the system is in manual mode, the lamp H1 must be ON and the lamp H2 must be OFF. Once the system is in manual mode, the switches can be used to engage/disengage the pumps and the mixer valve. The temperature transducers can measure the temperature between −25–400 °C and provide an unified electric signal of 0–10 V.

*Functional description 2 (automatic mode):* Pressing the S3 button will set the control system into automatic mode. When the system is in this mode, the lamp H2 must be ON and the lamp H1 must be OFF. When the agent temperature (TT1) is lower than 90 °C then

the PLC should close the bypass to direct the hot air towards the heat exchanger. When the agent temperature is higher than 90 °C, the bypass opens, and the hot air goes directly to the atmosphere. Once the bypass has been opened, the PLC should close it only when the TT1 has reached 80 °C. As a safety precaution the bypass is normally open. The mixer valve Y1 and the recirculation pump M1 ensure the recirculation of the agent through the heat exchanger and the buffer tank. If the temperature in the buffer tank drops below 80 °C (TT2) the mixer valve Y1 should commute to buffer tank and M1 should start. If the temperature reaches 85 °C then the mixer valve Y1 must commute to the recirculation system. As a safety precaution, when the temperature in the buffer vessel reaches 180 °C, the bypass must be opened, the valve Y1 must be closed and the pump M1 must be shutdown. Button S6 is used to drain water from the boiler. When the pressure of the water circuit drops below a set point then the PS1 (pressure switch) signal is enabled. The pump M2 should start in order to feed hot water into the circuit. After some time, the PS1 signal will be turned OFF and the pump M2 must be stopped. When the pump is running, the temperature in the boiler will decrease.

The system is able to switch between the manual and automatic mode in order to ensure a smooth operation. In manual mode, the user can activate various equipments without considering the safety measurements. In automatic mode, the heater will rise the temperature of the agent to a set constant. When the agent is hot enough, then the temperature of the water in the boiler can be influenced. The conceptual discrete state model of this process is listed in Figure 9.

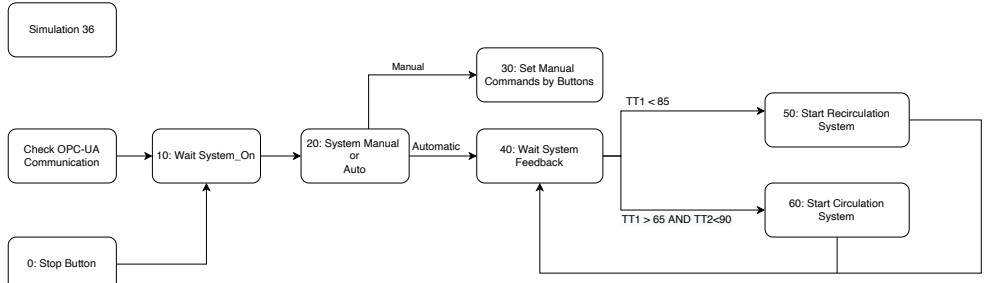

**Figure 9.** UML diagram—heat exchanger.

Being able to produce, store and process a large quantity of data is essential in order to detect patterns and apply a predictive maintenance strategy. Thus, in the current state of the industrial process, it is necessary to be able to integrate different equipments. The simulation of a smart metering system (Figure 10) showcases such functionality and allows to trainee to collect and interpret data from an industrial point of view.

*Functional description*: The system consists of three measuring lines. Each line is equipped with a flowmeter capable of monitoring the flow through the pipe and sending the available data via a Modbus communication. Each flowmeter has its own Modbus server with the corresponding parametrization: the first line Modbus Server has the IP address 192.168.1.3 and Port 502; the second line Modbus Server has the IP address 192.168.1.3 and Port 503; the third line Modbus Server has the IP address 192.168.1.3 and Port 504. The Start button will start the simulation of the data of the corresponding pipe line. The Stop button will end the simulation. The last two values set by the flowmeters are cumulative. Once a batch has started, the control system must start the monitoring. At the end of a batch, the total flow of oil of the current batch, the average temperature and the mean concentration must be stored in the PLC. The pressure transducers can measure the pressure between 1–8 bar and provide an unified electric signal of 0–10 V. A detailed listing of the Modbus registers used is presented in Table 2. The system can monitor three measurement lines, when the user starts the current batch. The data is stored into the PLC memory and waits for new commands. The application logic adheres to the diagram in Figure 11.

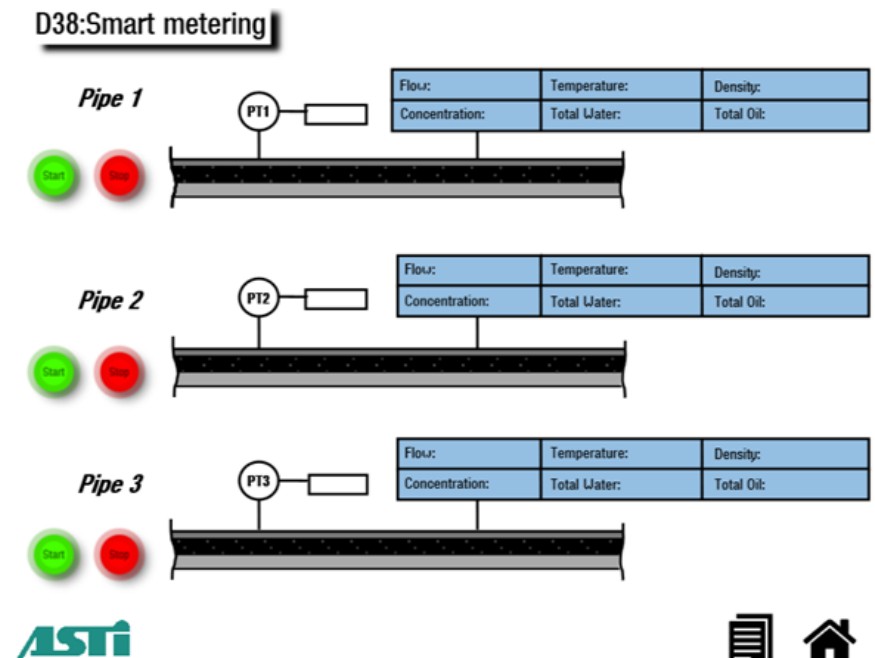

**Figure 10.** Smart metering simulation environment.

**Table 2.** Modbus map of each flowmeter.

| Number of Registers | Hold Registers | Type | Name | Range | Unit |
|:---:|:---:|:---:|:---:|:---:|:---:|
| 2 | 40,003 | Float | Flow | $0 \div 1000$ | $m^3/h$ |
| 2 | 40,005 | Float | Temperature | $-40 \div +85$ | $^{\circ}C$ |
| 2 | 40,007 | Float | Density | $750 \div 997$ | $kg/m^3$ |
| 2 | 40,009 | Float | Water concentration | $0 \div 100$ | % |
| 2 | 40,011 | Float | Total water | | $m^3$ |
| 2 | 40,013 | Float | Total oil | | t |

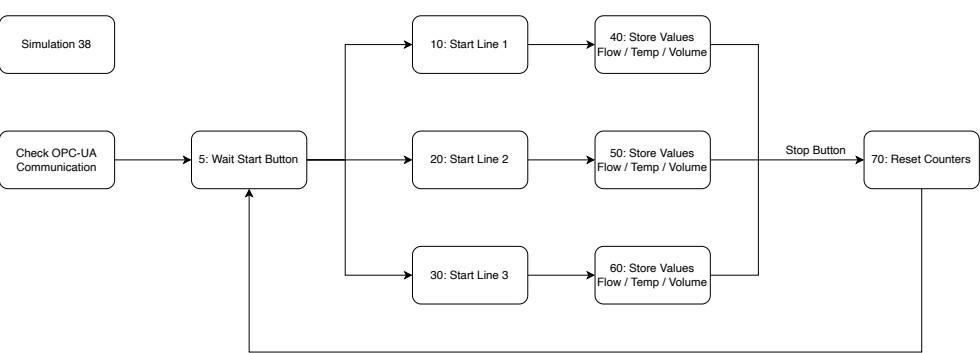

**Figure 11.** UML diagram—smart metering.

### *5.2. Implementation Aspects and Sandbox Deployment*

For OPC-UA development, the OPCFoundation.NetStandard.Opc.Ua C# library was used, which contains the OPC-UA reference implementation and is targeting the .NET Standard Library. This requires the allocation of the previously described Modbus register variables onto OPC-UA tags. We illustrate three code snippets for the dependencies inclusion, tag definition and tag update associated with the variables used by the process simulations.

**OPC-UA dependencies inclusion code snippet**

```
using Opc.Ua;
// Install-Package OPCFoundation.NetStandard.Opc.Ua
using Opc.Ua.Client;
using Opc.Ua.Configuration;
using System.Collections.Generic;
```

**OPC-UA tag definition code snippet**

```
TagList.Add("M1", new MainWindow.OPCUAClass.
TagClass("M1", "::AsGlobalPV:M1"));
TagList.Add("M2", new MainWindow.OPCUAClass.
TagClass("M2", "::AsGlobalPV:M2"));
```

**OPC-UA tag update code snippet**

```
MonitoredItem item = new MonitoredItem() { DisplayName = "Fault_M1",
StartNodeId = "ns=" + 6 + ";s=" + "::AsGlobalPV:Fault_M1" + "" };
                WriteValue valueToWrite = new WriteValue();
                valueToWrite.NodeId = item.StartNodeId;
                valueToWrite.AttributeId = item.AttributeId;
                valueToWrite.Value.Value = Convert.ToBoolean(0);
                valueToWrite.Value.StatusCode = StatusCodes.Good;
                valueToWrite.Value.ServerTimestamp = DateTime.MinValue;
                valueToWrite.Value.SourceTimestamp = DateTime.MinValue;
                WriteValueCollection valuesToWrite =
                 new WriteValueCollection();
                valuesToWrite.Add(valueToWrite);
                results = null;
                diagnosticInfos = null;
myOPCUAServer.OPCSession.Write(
                null,
                valuesToWrite,
                out results,
                out diagnosticInfos);
```

On the PLC side we use the OPC-UA server functionality provided with the simulated B&R PLC in ARSim. The process simulations act as OPC-UA clients to exchange data and commands with the control unit. The ARSimPLC project is built as a stand-alone application that runs in the background of the host machine and interfaces with the process simulations realized as C# Windows GUI applications. All the dependencies are built in the package and the solution does not require any development license for the automation software.

Once the dedicated Windows 2019 Server VM sandbox is provisioned, the executable files for each component along with the required dependencies. Once the tool is published, it can be accessed and operated by other users of the HUBCAP platform. The tool can also be shared with advanced users for additional configuration privileges.

Table 3 presents a snapshot of the resources required during operation by one of the process simulations controlled by the virtual PLC. It measures the cpu running seconds, the amount of data read and written to disk and to the input output system. The allocated random access memory and cpu number of cores are also listed. The virtual machine, named PRO-CPS in this case, can also access a shared storage space, where resources are pooled together with the other active VMs. At the administrator level, the VM can request more resources either statically or dynamically. Other options are provided by the sandboxing environment in the form of data export, e.g., saving outside the machine the results after running a certain model for offline analysis. Conversely, initial data can also

be uploaded to the sandbox to serve as preliminary resource to the running model or tool. These are provided as archives located at a specified path inside the VM.

**Table 3.** Simulation VM metrics.

| SBOX_ASSET | CPU_sec | RAM_gb | Cores | NET_IO_kb | DSK_RW_mb |
|---|---|---|---|---|---|
| Shared_Storage | 189 | 1 | 1 | 390,069 | 1241 |
| PRO-CPS | 27,356 | 8 | 4 | 27,352 | 6462 |

An example of the process simulator running in a HUBCAP sandboxed VM is provided in Figure 12a, and includes the VM management interface with the default Windows operating system. Figure 12b includes the running process simulation for the pumping station. The ARSim virtual PLC runtime is active in the background to monitor user input to the simulation UI and responds by generating commands through the linked industrial communication tags. Upon closing the active simulation program, the user is returned to a dedicated launcher application which allows the selection of a different simulation. The simulation ID is transferred to the PLC which also switches the active control logic which is preloaded in memory. In order to modify or create new control programs the full development environment must be installed in the VM. This is also the case when the user needs to modify the hardware configuration of the virtual automation system, for example to switch to a different CPU or add digital and analog input–output modules to accommodate a larger number or process variables and control signals.

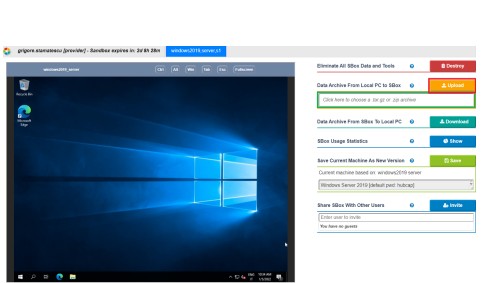
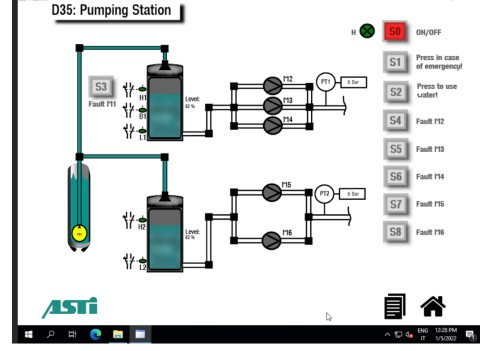

(**a**) Virtual Machine.　　　　　　　　　　　　　　(**b**) Software Application.

**Figure 12.** PROSIM PLC remote training and automation solution running in the cloud through a HUBCAP sandboxing environment VM.

## 6. Conclusions and Development Perspectives

The article presented and end-to-end solution for cloud-based, virtualized, remote industrial automation training platform that leverages the HUBCAP sandboxing environment for deployment and collaboration. The solution leverages several phases of development for an industrial process simulator whereby new Industry 4.0 technologies are integrated. Standardized industrial communication is an essential instrument that enables a scalable and replicable solution, which in our case is represented by OPC-UA integration. The main advances that allowed the virtualization of the developed PROSIM industrial process simulation into a cloud-enabled solution include:

- New simulations for the process industry including pumping systems, heat exchanger and smart metering applications that embed both the backend functionality with OPC-UA client libraries and the frontend user interface; a modelling stage using has been carried out for a proper MBD methodology;

- Virtual PLC controller running in the background of the host machine; this leverages existing automation services and includes the control logic for each of the simulated processes as well as supporting the local OPC-UA server functionality;
- Deployment of the cloud tool in the HUBCAP sandboxing platform through provisioning and configuration of a dedicated virtual machine with required dependencies; dynamic adjustment of the available VM resources ensures the scalability of the training solution through multiple simulations in a single VM or multiple VMs running in parallel on the platform to assure concurrent access to the software.

Several key lessons learned over the development of the fourth, cloud-based, PROSIM solution are highlighted. The need for a robust cloud platform that provides ease-of-use to a heterogenous audience, composed of students, technicians and engineers, is considered highly important. Challenges in the access and performance of the virtualized environment can discourage the student and distract them from the key concepts and practical skills that should be the focus of the training. With regard to the development of new process simulations that can run in the cloud, these should aim for a suitable balance between realistic models of industrial automation applications, covering electrical diagrams, control narratives and i/o lists, as well as rich visual feedback through quality diagrams, objects and animations. Finally, the simulation solution for automation and PLC training should constantly be improved to keep up with the pace and anticipate the development of new industrial technologies—sensors, controllers, communication protocols and software components—while maintaining support for existing and legacy technologies to account for the extended upgrade cycles of industrial automation applications.

The current limitations of this work are mainly related to the integration of open application programming interfaces (APIs) that allow interaction with external systems and real laboratory equipment. This could improve the applicability of the solution to a wider range of training scenarios.

Validation of the system with end users in professional training courses is foreseen which will lead to continuous improvement. This can be best achieved in hybrid mode, where the trainees gather experience working on the virtual system under professional guidance, followed by hands-on laboratory sessions on the real equipment. The simulation can also be operated in parallel to the real equipment for grasping modelling limitations and ideal behavior compared to potential faulty operation that is encountered in practice. Moreover, extending the number, type and complexity of the existing simulation towards fully functional digital-twin models with discretized continuous system dynamics will be carried out.

**Author Contributions:** Conceptualization, G.S. and V.M.; methodology, V.M. and S.R.; software, S.R.; validation, M.N. and D.C.; formal analysis, S.R., G.S.; writing—original draft preparation, G.S., S.R. and M.N.; writing—review and editing, D.C. and S.R.; visualization, V.M.; supervision, G.S. and M.N. All authors have read and agreed to the published version of the manuscript.

**Funding:** This work was is partially supported by the HUBCAP Innovation Action funded by the European Commission Horizon 2020 Programme under Grant Agreement 872698, HUBCAP Call 2.1. Experiment, Subgrant no. HUBCAP-OC2.1-2020/1499086, PROSIM Cyber-Physical Platform for Application Development and Training in the Process Industries (PRO-CPS). Financial support from the University Politehnica of Bucharest is gratefully acknowledged.

**Conflicts of Interest:** The authors declare no conflict of interest.

## Abbreviations

The following abbreviations are used in this manuscript:

| | |
|---|---|
| AR | Augmented Reality |
| CPS | Cyberphysical Systems |
| FBD | Function Block Diagram |
| HMI | Human Machine Interface |
| IIOT | Industrial Internet of Things |

| JSON | JavaScript Object Notation |
| MBD | Model-based Design |
| OPC-UA | Open Platform Communications-Unified Architecture |
| PLC | Programmable Logic Controller |
| UI | User Interface |
| VM | Virtual Machine |
| VR | Virtual Reality |

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
