# Peer review of "PROSIM in the Cloud: Remote Automation Training Platform with Virtualized Infrastructure"

_applsci, doi:10.3390/app12063038_

Round 1

Reviewer 1 Report

The research topic is interesting but Extensive editing of English language and style required

Author Response

We would like to thank the reviewer for studying the manuscript and providing valuable comments. Please see attachment for our response.

Reviewer 2 Report

The topic of modern automation technology under the scope of Industry 4.0 and the associated training approaches fits the scope of the Journal. The manuscript requires some extra efforts to improve its quality and presentation for the prestigious journal Applied Sciences. A set of comments are expounded hereafter.

- The manuscript is, in general, well organized and well written. However, there are some issues regarding the format of the document, as commented below.

Acronyms must be decomposed the first time that they are used in the paper, even, in the Abstract. For example, PLC, OPC, etc. can be found in the Abstract.

The acronym PLC is decomposed in line 501 whereas in line 24 it is not used. This issue must be solved for a proper presentation.

Modbus and ModBus are both found in the manuscript. Only one form must be used.

Figure captions and titles of tables lack the terminal period (punctuation).

Concerning the references, the format must be revised to follow the template concerning some aspects like the abbreviated name of the journals, quotation marks (not required), etc.

- About the content of the manuscript, it covers a very interesting topic. The comments after a careful revision are the following:

It is suggested to include as keyword “Industry 4.0”. This will enhance the visibility of the paper.

In a similar sense, the role of PLC in Industry 4.0 could be highlighted by means of recent papers, for example:

-The PLC as a Smart Service in Industry 4.0 Production Systems. Appl. Sci. 2019, https://doi.org/10.3390/app9183815

The authors could mention the fact that PLCs have an expected lifespan of decades, which highlights the interest of proper automation training.

The proposal is well described concerning hardware, software and communications.

Using Modbus TCP is a good choice given its widespread application in industrial facilities.

Concerning the operating system, the reported work uses MS Windows. As the authors know, the OPC-UA protocol can be used also for Linux-based systems. This positive feature could be mentioned in order to enhance the description of such protocol as well as to make the proposal more applicable.

The main limitations of the work could be briefly mentioned in the Conclusions.

Author Response

(The authors gave the same response as above.)

Reviewer 3 Report

The paper describes a learning and training environment for automatic control. 

Unfortunately the paper focus very much on the history and the current system developed. It discusses in very detail the technical details.  

Missing is a discussion of the state-of-the-art. In particular no e-learning environments others that the ones compared are discussed. Also no references to training in automatic control is made. 

For the current content, it should be focused on a higher level of abstraction.  

Author Response

(The authors gave the same response as above.)

Round 2

Reviewer 3 Report

Thank you for highlighting how my comments have been addressed.

Response 3.1: Instead of going into details, that allow me to make an exact copy, I would have liked to see why changes are made (shortcomings of previous versions). Highlighting also the advantages of the current improvements. 

Response 3.2. An introduction introduces the reader to the problem at hand. Its not an state-of-the-art section. Please add a separate section for that. When doing so, you will realize that this state of the art review is very short. 

Response 3.3. ok.

Still I am missing an extended lessons learned part in the conclusions section. 

Author Response

We would like to thank the reviewer for the time and patience in considering our work. Please see the attachment for detailed responses to the raised comments.
